# Elucidating Patterns in the Occurrence of Threatened Ground-Dwelling Marsupials Using Camera-Traps

**DOI:** 10.3390/ani9110913

**Published:** 2019-11-03

**Authors:** Andrew W. Claridge, David J. Paull, Dustin J. Welbourne

**Affiliations:** 1Office of Environment and Heritage, National Parks and Wildlife Service, Nature Conservation Section, Queanbeyan, NSW 2620, Australia; 2School of Science, University of New South Wales, Canberra, ACT 2601, Australia; dpaull@adfa.edu.au (D.J.P.); dustin.welbourne@ufl.edu (D.J.W.); 3NSW Department of Primary Industries, Vertebrate Pest Research Unit, Queanbeyan, NSW 2620, Australia; 4Department of Wildlife Ecology and Conservation, University of Florida, Gainesville, FL 32603, USA

**Keywords:** bandicoot, camera-traps, detection, habitat, mammal, marsupial, monitoring, occupancy, potoroo, rat-kangaroo, threatened

## Abstract

**Simple Summary:**

Being able to effectively monitor the continued plight of highly vulnerable animals against management efforts over time is critical for their conservation. In south-eastern New South Wales, Australia, we used a camera trapping array to collect baseline information about patterns of occurrence of three threatened native ground-dwelling marsupials of conservation interest: the long-nosed bandicoot (*Perameles nasuta*), long-nosed potoroo (*Potorous tridactylus*) and southern brown bandicoot (*Isoodon obesulus*). Over a four-year period, detections of the two bandicoots were more erratic and less predictable than that of the potoroo, resulting in higher uncertainty about occupancy estimates and adequacy of sampling effort. The detection probability of each bandicoot species and that of the potoroo differed variously with structural complexity of vegetation. Detection probability of the southern brown bandicoot was highest where ground cover was most dense and shrub cover most open. The reverse pattern was found for the long-nosed bandicoot. Finally, the detection probability of the long-nosed potoroo was highest where ground and shrub cover was densest. Future camera trapping monitoring efforts need to take better account of these nuances and be flexible to including additional sampling for at least the two bandicoots. In short, when it comes to monitoring approach, one size doesn’t fit all.

**Abstract:**

Establishing trends in endangered fauna against management efforts is a key but often challenging enterprise. Camera-traps offer a new and literal window into monitoring many different mammalian species. Getting it right demands seeking baseline information about how often target species interact with these devices, prior to setting a long-term monitoring strategy. We used a camera-trap array to collect detection data on three species of threatened ground-dwelling marsupials in south-eastern mainland Australia. Over a four-year period, occupancy estimates for two species of bandicoot (southern brown bandicoot *Isoodon obesulus* and long-nosed bandicoot *Perameles nasuta*) and a single species of rat-kangaroo (long-nosed potoroo *Potorous tridatylus*) were generated. These estimates were variously robust depending on visitation history, but nevertheless indicated persistence of these rare and otherwise under threat species. Detection probability for each species differed between study areas, type of management and with complexity of ground and shrub vegetation cover. The relationship between detection and vegetation structure dictated that survey effort was only robust where conditions were optimal for a given species. Outside of that further survey effort would be required to have confidence in survey outcome. In the future this would demand a different sampling strategy, be that through lengthening survey time or adding additional camera units at sites.

## 1. Introduction

Camera-traps are playing an ever-prominent role in survey for and monitoring of wildlife around the globe, including that for cryptic and endangered species i.e., [1,2,3,4,5]. Unlike traditional survey and research methods, camera-traps offer logistical and ethical benefits. These include the ability to set them for relatively long periods of time without checking and allowing wildlife to move freely about them while they census the environment [6]. There is also increasing evidence that camera-traps increase detection rates of target fauna, particularly when set in association with lures or in parts of the landscape naturally used by animals [5,7,8,9,10]. In the Australian context, camera-traps are a relatively new addition to the suite of surveillance techniques being used to census wildlife. Notwithstanding, there has already been a large body of work reported on and published, particularly in the last decade or so [11]. For the most part this work has focused on various aspects of detection, survey and monitoring of mammals, and to a lesser extent birds and then reptiles [12]. That said, recent modifications to camera traps and arrangements for setting them up in the field is improving their utility for censusing the latter group of fauna [13,14].

Australia has many well documented fauna-related conservation management challenges. Among these is trying to curb an appalling and ongoing history of loss of so called critical-weight-range mammals [15]. Since European human settlement, this group of mammals, which fall between 35–5500 g in body weight, have been particularly extinction prone due to a combination of factors including land clearing, inappropriate fire regimes and introduced predators such as foxes (*Vulpes vulpes*) and feral cats (*Felis catus*) [15,16]. These patterns have been acute among ground-dwelling mammals in arid and semi-arid areas of the continent during the past 150 years or so [17]. Declines are still in evidence and even continue to emerge elsewhere in areas where these native fauna were previously considered to be stable i.e., [18]. Those taxa worst hit by catastrophic decline include marsupial bandicoots (Order Peramelemorphia) and rat-kangaroos, including potoroos (Family Potoroidae) [19]. Keeping an ongoing watch on these marsupials is a priority, particularly as management regimes attempt to improve their conservation status through different interventions.

Bandicoots and potoroos can be notoriously difficult to census, making it difficult to infer ongoing trends with any level of confidence. Direct techniques such as live-trapping can be grossly inefficient, with very low trap success [20]. As a replacement method, hair-sampling tunnels became popular for censusing these marsupials in the mid-1980’s through to the mid-2000’s i.e., [21]. They were initially praised for their ethical advantages over live-trapping as they only required individual animals to leave hair behind on double-sided adhesive tape, rather than be entrapped. Hair tunnels were also found capable of censusing many species, potentially all at the same time, depending on design [22]. Despite these supposed advantages, the ability of hair tunnels to record the presence of bandicoots and potoroos was found to be spectacularly poor when compared to background sign of the same animals, such as forage-diggings, at the same sites [23,24].

Indirect sampling methods have also been used to survey and monitor bandicoots and potoroos, but they too have their limitations. Claridge and Barry [19] used forage-diggings of these animals to discern patterns in their occurrence at a landscape-scale across south-eastern mainland Australia. While these forage-diggings were sufficient to indicate presence and absence of the target species, they were not capable of telling species apart. Signs left by bandicoots could have been made by either or both of two bandicoot species and, likewise, signs left by potoroos by either or both of two species. Similarly, Claridge et al. [25] and Arthur et al. [26] used tracks in sand plots to examine patterns in the occurrence of bandicoots and potoroos in relation to habitat, time since fire, climate and predator control. Despite being able to develop explanatory models of their occurrence and relative abundance, individual bandicoot species could not be identified based on their tracks. This limits interpretation of subsequent trends and says little about how populations of individual species might be changing over time.

In contrast to the low efficacy of direct techniques such as live-trapping and hair-sampling, and the poor discriminatory power of indirect signs such as forage-diggings and tracks, camera-traps offer a potential and literal window into observing change in bandicoot and potoroo populations. In earlier work, Claridge et al. [27] provided firm evidence of the ability of camera-traps to discriminate bandicoot and potoroo species from images and at a site-level record them at a rate consistent with background evidence such as the presence of forage-diggings. Here, we extend this earlier work by exploring patterns in the occurrence of three threatened ground-dwelling marsupials across a broad coastal landscape in south-eastern New South Wales, Australia: the long-nosed bandicoot (*Perameles nasuta*), long-nosed potoroo (*Potorous tridactylus*) and southern brown bandicoot (*Isoodon obesulus*) (Figure 1). The latter two species are listed as endangered under Australian Commonwealth and State legislation and monitoring their fate over time is a high priority. Using camera-traps we examined how occupancy of these species changed through time and evaluated the effectiveness of camera-traps in different habitats. The overall purpose of this preliminary study, conducted between 2015 and 2018, was to help set a foundation for ongoing monitoring of this remnant guild of threatened ground-dwelling marsupials.

## 2. Materials and Methods

### 2.1. Study Area

Our camera-trap monitoring study occurred within the southern section of Ben Boyd National Park (Ben Boyd), and the northern half of adjacent Nadgee Nature Reserve (Nadgee), south of the coastal township of Eden in south-eastern New South Wales, Australia (Figure 2). Earlier live-trapping work there showed that bandicoots and potoroos, the focus of our work, were present although trap success was extremely low: around 0.5 captures per 100 trap-nights for long-nosed bandicoot (hereafter *Perameles*) and southern brown bandicoot (*Isoodon*), and 2 captures per 100 trap-nights for long-nosed potoroos (*Potorous*) (NSW National Parks and Wildlife Service, unpublished data). The main features of both study areas, include climate, geology, topography and major vegetation types, have been described in detail in a related publication [27], as well as reserve management plan [28]. Briefly, most of Ben Boyd lies <160 m above sea level and the landscape is gentle to undulating. The geology is predominantly comprised metamorphosed Devonian sediments and associated sandstones, with soils largely being mostly free-draining sandy sediments with some Tertiary dunes. The climate is mild, with average mean maximum temperatures around 18 °C and average mean minimum temperatures a little over 12 °C, with summer highs and winter lows. Average annual rainfall is ca. 750 mm, which is generally evenly distributed throughout the year (www.bom.gov.au/climate/averages/tables/cw_069055.shtml). The vegetation is predominantly open forest and woodland dominated by a range of eucalypt species, woody midstorey shrubs and variable ground cover of rushes, ferns and sedges. Elsewhere across this park, heathlands occur, particularly directly on the coastline. Prior to becoming a conservation reserve Ben Boyd was subject to broadscale logging and prescribed burning, particularly during the 1970’s. The last significant wildfire was in 1980/1. Nadgee lies immediately to the south of Ben Boyd but is separated by the ocean and associated estuaries. Spanning 21,000 ha, Nadgee shares it’s climatic and geological features with Ben Boyd and features dissected low tablelands, coastal plain, estuaries, coastal lagoons and beaches [28]. Open forests containing a variety of eucalyptus species are the most widespread vegetation type there, occupying ridges, hills and well-drained coastal areas. Coastal scrubs and heaths occur elsewhere, behind beaches and headlands. An extensive, high intensity wildfire burnt much of the reserve in 1972, and again in the northern and central parts in 1980.

Ben Boyd has been subject to intensive and widespread control efforts for introduced foxes (*Vulpes vulpes*) through 1080 baiting for well over a decade, aimed at reducing predation pressure on bandicoots and potoroos [27]. These long-term works on-park are supplemented by a similar program on adjacent State forest tenure further inland, creating a larger buffer about the study area. In addition, leg-hold trapping of feral cats (*Felis catus*) is also undertaken intermittently across Ben Boyd, as resources permit. This latter control effort commenced in 2017. In contrast, Nadgee remains unbaited for foxes and free from trapping of feral cats.

### 2.2. Camera-Trap System

The camera system we used in Ben Boyd and Nadgee has been previously described in a related paper [24]. In brief, our survey and monitoring efforts were conducted using Reconyx™ PC90 camera-traps (hereafter PC90) (Reconyx Inc, Holmen, WI, USA). For a few years preceding our work the PC90 was a leading model camera-trap, renowned for its fast trigger speeds (1/5 s) and covert (low-glo) night-time illumination system. Though now superseded, it still shares similar attributes to newer model camera-traps, not only relating to responsiveness and illumination, but also image quality (3.1 megapixel color images during daylight and 3.1 megapixel monochrome (black and white) images at night). Either way, the PC90 is still a capable camera-trap. To improve species identification, we set camera-traps to “rapidfire mode”, whereby a sequence of 10 JPEG images were taken for each time the unit was triggered, 24 h a day. Images taken by each camera-trap were stored on-board on a 2 GB SanDisk Ultra Compact Flash (CF) card (SanDisk Manufacturing Limited, Dublin, Ireland). In the field, each camera-trap was affixed to a 1 m long stainless-steel rod via a Thunderbolt mounting block (Reconyx Inc, Holmen, WI, USA) [24]. Camera-traps were set approximately 10–20 cm above the soil-litter interface and 1.8–2.0 m horizontal distance from a bait holder, held into the ground using a steel peg. The bait holder comprised a 50 mm PVC vent cowl (Vinidex Pty Ltd., Bohle, Queensland, Australia), within which a standard mixture of peanut butter and rolled oats was placed. At the end of each deployment, which lasted between 30–40 days, CF cards were retrieved from each camera and image files screened on a computer. Wherever possible, animals were identified from these files to species level, which was generally not problematic.

### 2.3. Camera-Trap Layout

Ben Boyd and to a greater extent, Nadgee, are characterized by limited access and sometimes extensive dense vegetation. This severely hampers being able to deploy camera traps in any randomized or grid-based pattern as it too logistically demanding. Instead, we used the existing vehicular track and walking trail network as a base from which to select deployment sites. To do this, a random seed point along a vehicle trail in the far northern end of Ben Boyd was selected, and subsequent sample points chosen every 500 m from that point along all existing vehicle trails as well as major walking trails, south through Nadgee. The minimum 500 m spacing between camera traps was chosen to represent a suitable distance apart to largely avoid sampling the same individuals of the target species, based on known home range sizes, maintaining a level of independence in sampling outcomes. From each sample point, camera traps were then set-up to between 20 and 100 m off the track or walking trail into the surrounding vegetation. Using this spacing 84 sites were located for long-term annual camera trapping in Ben Boyd and 84 sites in Nadgee. Final site selection was further based on ensuring that there was proportional representation of sites across each of the major vegetation classes mapped for both reserves (Figure 3). Where vegetation classes were under-sampled sites were re-allocated. In this respect the sampling regime was not biased to preferred habitat types for the two bandicoots and the potoroo; instead, camera trap sites were uniformly spaced.

Due to limited stock of PC90’s, sampling across the 168 sites was split into two sessions, typically between May and September each year. For the first session, half of the sites (n = 42) in the southern end of Ben Boyd were sampled and half (n = 42) in the northern end of Nadgee were camera trapped. For the second session, the other half of the sites in the northern end of Ben Boyd and southern end of Nadgee were similarly sampled. Camera trapping results from both sessions were then combined to provide an estimate of occupancy (see below) for each of the three target species for that year. From one year to the next, the same sites were re-sampled at roughly the same time of year. Given the length of time camera traps were set for (30–40 days) and that all work was done in the cooler months of the year, any seasonal effects in detectability are considered minimal. None of the target species we were working with are known to display significant movements away from existing home ranges or otherwise alter their activity patterns grossly at a seasonal time scale. They also breed continuously throughout the year and so reproductive condition does not likely effect detectability either.

### 2.4. Habitat Complexity Scoring

At each camera-trap site we recorded structural features of living and non-living vegetation, and other habitat features, against which to examine patterns in detection probabilities of bandicoots and potoroos. To do this we used a form of habitat complexity scoring system first developed by Newsome and Catling [29], as indicated in Table 1. Our system differed from their original version by separating scores for woody debris and rocks and adding an additional score for leaf litter layer. Scores for each habitat attribute were estimated over a 50 m × 20 m rectangular plot at each camera trap site, with the plot aligned along the dominant contour. Of these attributes, two were of particular interest in further analyses, ground and shrub cover, as previous studies had shown they influenced the occurrent of bandicoots and potoroos [19].

### 2.5. Data Analyses

To answer the questions of the study two primary types of analyses were performed. First, to examine how occurrence of the target species varied through time in the study area, we estimated single season and multi-season occupancy as described by Mackenzie et al. [30]. Two single-season models for each species were evaluated: a null model, where occupancy and detection probability (ψ(.) p(.)) were considered constant across the study area; and, a model where occupancy varied based on the area (Ben Boyd or Nadgee) while detection probability remained constant (ψ(area) p(.)). Given the detection histories spanned multiple years (2015–2018), four multi-season models were also evaluated for each species: (i) Initial occupancy, colonization, extinction, and detection probability were assumed to be constant (ψ(.) γ(.) ε(.) p(.)); (ii) initial occupancy varied by area, but colonization, extinction, and detection probability were constant (ψ(area) γ(.) ε(.) p(.)); (iii) initial occupancy varied by area and colonization and extinction varied by year, while detection probability was constant (ψ(area) γ(year) ε(year) p(.)); and, (iv) initial occupancy and detection probability were constant but colonization and extinction varied by year (ψ(.) γ(year) ε(year) p(.)). Occupancy estimates for t + 1 years were calculated from colonization and extinction estimates using a parametric bootstrap approach (1000 iterations) from the following:ψ_(t + 1)=ψ_t (1 − ε_t)+(1 − ψ_t)γ_t(1)

Second, to examine camera-trap effectiveness throughout the study area and determine the adequacy of current survey effort, we first examined daily detection probability using Bayesian methods with vague priors. Since credible intervals in the Bayesian approach represent the degree of confidence that the parameter of interest is within said interval (an interpretation that cannot be made with the frequentist approach), we then used the estimates to calculate the number of survey days required to detect a target species. Estimates of daily detection probability were first conditioned upon occupancy; that is, if a species was not detected at a given site during a season, that site was removed from the analysis for said species. For a given species, detections (*d*) on the *i^th^* day at the *j^th^* station were assumed Bernoulli distributed:(2)$$d_{ij}\;\sim \;Bern\left(\delta_{j}\right)$$
and detection probability (*δ*) was estimated using a logistic model (Equation (3)):(3)$$\mathrm{logit}\left(\delta_{j}\right)=\ln\left(\frac{\delta_{j}}{1-\delta_{j}}\right)=\beta_{0}\;+\beta_{1}S_{j}+\beta_{2}H_{gj}+\beta_{3}Y_{k}$$

Baiting (*S_j_*), habitat complexity (*H_gj_*; ground cover and shrub cover scores), and year (*Y_k_*; 2015–2017) were covariates in the fully-specified model. Vague normal priors were used for *β_0_* through *β_3_* with *mean* = 0 and *precision* = 1^−6^:(4)$$\beta_{k}\;\sim \;N\left(mean,\;precision\right)$$

Five models were compared for each species. A null model, whereby baiting, habitat, and year were assumed to have a constant effect. A bait only model, whereby whether baiting occurred at the site affected detection probability. A habitat only model whereby only habitat (ground and shrub cover) affects detection probability. A habitat and bait model, whereby the habitat and whether baiting occurred affected detection probability. Lastly, a fully-specified model whereby detection probability was a function of habitat, baiting, and year.

The minimum and maximum daily detection probabilities, which represent a best- and worst-case scenarios for detecting a species, were used to calculate the number of survey days required to be 95% confident that non-detection of a species could be interpreted as its absence. Using the estimated daily detection probability (*δ*), the number of trap days (*n*) that would be required to be 95% certain (*α*) that a species was not present at a site was calculated from:(5)$$n=\frac{\log\left(1-\alpha \right)}{\log\left(1-\delta \right)}$$

Model fit and complexity for both occupancy and Bayesian analyses were measured using Akaike information criterion (AIC) [31] and deviance information criterion (DIC) [32] (respectively). Determining the most likely model candidate was done by examining the change in AIC (ΔAIC) or DIC (ΔDIC) between a given model and the model with the lowest value. Table 2 outlines the levels of support for models given their ΔAIC or ΔDIC value. Where multiple models had substantial support, estimates were generated using weighted model-averaging.

All analyses and data processing were conducted using R Version 3.5.3 [34]. We used the package Unmarked [35] to perform single- and multi-season occupancy estimation. To perform Bayesian analyses, the package R2OpenBUGS [36] was used in R to call OpenBUGS (version 3.2.3) (MRC Biostatistics Unit, Cambridge, UK) [32]. OpenBUGS software is an open source environment used to run Bayesian analyses with Markov Chain Monte Carlo (MCMC) techniques [32]. Bayesian models were run using two MCMC chains. The R package CODA (Convergence Diagnosis and Output Analysis) was used to assess chain convergence [37]. Convergence was confirmed by visually examining convergence and autocorrelation plots, and by using Gelman and Rubin’s convergence diagnostic test to ensure shrinkage of parameter estimates were <1.05 [38]. Iterations for Bayesian models varied depending on convergence, and half of the iterations were used as burn-in.

### 2.6. Ethics and Wildlife Licensing Permits

Field research was conducted under the provisions of a NSW National Parks and Wildlife Service Scientific Investigation Licence (10018) and an approval from the Office of Environment and Heritage Animal Care and Ethics Committee (980315/01).

## 3. Results

### 3.1. General Trends

Camera-trapping surveys were conducted across the 168 sites over four consecutive years. Due to camera-trap malfunctions and theft, not all sites were surveyed continuously for the 30–40 days during each year. This did not present a problem for the occupancy modelling process. Using a 30-day survey period resulted in 19,350 camera-trap days being effected, whereas the 40-day survey period resulted in 25,800 camera-trap days. Although detection frequency of *Isoodon* and *Perameles* increased from year to year, *Potorous* was detected more frequently overall (Table 3). Both *Perameles* and *Potorous* were detected more frequently in Nadgee in each year of the study, whereas *Isoodon* was detected more frequently in Ben Boyd (Figure 4).

### 3.2. Single Season Occupancy

Two single-season occupancy models were evaluated for each species in each year. The first, both occupancy and detection probability (*ψ*(.) *p*(.)) were considered constant, and the second, occupancy varied by the area while detection probability was constant (*ψ*(area) *p*(.)) (Table 4 and Table 5). Occupancy predictions based on the 40-day survey period were like predictions based on the 30-day survey period for all species, except in one instance. The 2017 predictions for *Perameles* in Nadgee were somewhat higher with the 40-day survey period than the 30-day survey period (Figure 5). This may suggest that closure assumptions of the 40-day time frame are not always met for this species due to its eclectic use of habitat. Except in the prior case, occupancy predictions for each species were less than 50% in both areas. Occupancy for *Isoodon* was considerably higher in Ben Boyd than Nadgee, a trend that was consistent throughout the study period. Although *Potorous* generally exhibited higher occupancy in Nadgee for 2015–2017, in 2018 there appeared to be no difference in occupancy between areas. 

### 3.3. Multi-Season Occupancy

The both *Isoodon* and *Potorous* the models with considerable support included area as a factor in estimating initial occupancy, with colonization, extinction and detection probabilities constant (Table 6). For *Perameles* the model with most support did not include covariates, yet all models had considerable support. Although not perfectly identical to single-season occupancy estimates, the multi-season derived occupancy estimates showed similar trends (Figure 6). There was little difference in occupancy between Ben Boyd and Nadgee for *Potorous*. Given model selection for *Perameles*, occupancy estimates did not vary between areas in each year. Nevertheless, the probability of occupancy of *Perameles* increased from year to year. Probability of colonisation (γ) and extinction (ε) were higher for *Perameles* than for either of the other species examined (Figure 7).

### 3.4. Detection Probability & Habitat Complexity

The model of best fit for each of the target species included whether the site was baited or not (i.e., Ben Boyd or Nadgee), the nature of the ground and shrub vegetation cover, and year as predictors (Table 7). Detection probability appeared higher in Ben Boyd for *Isoodon*, but higher in Nadgee for *Perameles* and *Potorous* (Figure 8). Vegetation affected detection probability differently for each target species. For *Isoodon*, detection probability increased as ground cover increased but shrub cover decreased. For *Perameles*, detection probability increased as ground cover decreased but shrub cover increased. Lastly, for *Potorous* detection probability increased as both ground and shrub cover increased.

### 3.5. Survey Effort

The lowest and highest detection probabilities were used to calculate the number of survey days required at a site to be 95% certain that non-detection of a species could be interpreted as its absence. For 40-day survey periods, only *Isoodon* and *Potorous* were effectively detected in the best-case scenario (Figure 9). In the best-case scenario for *Perameles*, although the lower bound and mean of the survey days required for effective detection fell within approximately 40 days, the upper bound extended to approximately 70 days. For all target species, the number of survey days required for effective detection fell well beyond a 40-day survey period.

## 4. Discussion

### 4.1. Utility of Camera-Traps

Endangered fauna often provide unique challenges when it comes to obtaining robust monitoring information to track their ongoing status [39]. Finding suitably efficient sampling techniques can be highly problematic [20]. In Australian landscapes threatened ground-dwelling marsupials exemplify this dilemma. Traditional methods for directly censusing these cryptic fauna, like live-trapping and hair sampling are logistically demanding, sometimes ethically questionable and above-all-else largely inefficient [23,24]. Furthermore, indirect survey techniques such as recording forage-diggings or tracks, are not necessarily species-specific, leading to ambiguity in outcomes [25,26]. Here, we used camera-traps to attempt to monitor three such marsupials: the long-nosed bandicoot (*Perameles*), long-nosed potoroo (*Potorous*) and southern brown bandicoot (*Isoodon*). We assessed detection rates of these species over a four-year period, in relation to duration of survey across two major study sites. One study site, Ben Boyd, was subject to intensive control of introduced predators (red fox and feral cat), while the other, Nadgee, was not treated in this way.

Detection frequency of all three-species increased across both study sites over time and with increasing survey duration from 30 to 40 days. *Potorous* was more detectable than either bandicoot species, particularly at Nadgee where there was no control of introduced predators. *Perameles* was similarly more detectable there. These patterns likely reflect the higher availability of optimal habitat for both species in Nadgee. In contrast, the detectability of *Isoodon* was higher at Ben Boyd, where foxes and cats were subject to control. Other studies have shown the benefit of introduced predator control for native marsupial species, with variable responses in time and space, meaning the outcomes are not always predictable i.e., [40].

When detection frequency data for the two bandicoots and *Potorous* were converted into single-season and multi-season occupancy models, similar as well as some slightly different trends were observed. Overall, extending the duration of camera-trap survey period from 30 to 40 days made little statistical difference to the estimates in single-season models across each of the four years of sampling. The single exception to this pattern was for *Perameles* at Nadgee, during 2017. Thus, for the most part it would appear the increases in detection frequency of animals were within existing sites rather than across sites. Otherwise, the occupancy estimates for *Isoodon* were consistently higher in Ben Boyd than for Nadgee, ranging from around one in five camera-trap sites to more than one-in three camera trap sites, compared to somewhere around one in ten camera-trap sites. For *Perameles* and *Potorous*, occupancy estimates were broadly similar between Ben Boyd and Nadgee across the four years, with *Potorous* being detected at up to four in ten camera-trap sites. These rates far exceed the detection success reported in live-trapping and hair-sampling studies of the same species [23,24,39].

Multi-season occupancy estimates were mostly parallel to those of single-season estimates, with little difference for *Perameles* and *Potorous* between Ben Boyd and Nadgee. However, there seemed to be an overall increase in occupancy of *Perameles* across time, irrespective of site, perhaps indicative of a broader population increase. Multi-season occupancy estimates also indicated a difference for *Isoodon* between Ben Boyd and Nadgee, with higher estimates in the former, although the difference diminished over time. Across the years, *Isoodon* and *Potorous* displayed a higher fidelity to camera-trap sites than *Perameles*, in so far as the turnover rate of previously occupied to newly occupied sites was much lower. This pattern could be an artefact of differences in the habitat preferences among these species (see below). The trend also reinforces the view that *Perameles* is an enigmatic species that is difficult to sample effectively [23,40].

### 4.2. Habitat Complexity and Detectability

Habitat structure was found to influence detection probability of each of the bandicoot species and *Potorous*, but in slightly different ways. Vegetation cover has previously been found to influence this guild of marsupials at landscape and site-specific scales, with denser vegetation mostly found to increase occupancy and relative activity i.e., [19,26,41]. Of itself this pattern should come as no surprise, given dense cover likely affords protection from mammalian predators, at least some of which are known to predate on these native marsupials [25,42,43]. The combination of camera-trapping and habitat complexity scoring used in the current study shows a slightly more nuanced pattern: *Isoodon* were detected more often at sites with a dense ground cover of vegetation, with a more open shrub cover above. This contrasted with *Perameles*, which was detected more often where ground cover was open and shrub cover dense, and *Potorous*, detected more frequently where ground and shrub cover was densest. These relationships suggest a degree of habitat partitioning among three similarly-sized ground-dwelling marsupials, allowing them to co-exist within the same general landscape. This finding also highlights the importance of maintaining vegetation in a range of structural conditions to preserve a diversity of native fauna [41,44].

From a monitoring point of view, the differences in detectability of bandicoots and potoroos with respect to habitat complexity presents an experimental challenge. Our modelling of effective survey lengths implies that the current duration of effort (40 days) was only sufficient for *Isoodon* and *Potorous* where habitat structure was most favorable for those species, respectively. Intuitively this makes sense as species should be more likely to be detected in favored habitat. However, outside of sites with optimal structure there was still uncertainty about survey outcome: In short, lack of evidence of these species cannot be considered absences with total confidence. For *Perameles* the situation was even poorer, with low detection probabilities across the board with respect to measures of ground and shrub cover. This resulted in survey outcomes with low confidence. This provides further evidence of the enigmatic nature of this latter species.

Detections of each of the target species can be increased by lengthening survey period. However, this may invalidate assumptions of site closure that the occupancy modelling used here demands [30]. Alternatively, increasing the number of camera traps per sampling site may likely improve detection probability as the area of habitat effectively surveyed at any given site is increased [45,46]. While possible to do, this would impact on the existing program as there are only a finite set of cameras in the pool available for the monitoring work. To address the deficiency would require a different allocation of cameras to sites, potentially slowing down the rate at which the overall set of sites get sampled. The rate at which our target species interact with camera-traps might also be improved by seeking alternative lure types. To-date there has been some exploration of different attractants for bandicoots and potoroos, including the use of invertebrates and oils with food additives. None of this work has so far resulted in an attractant that has been found to be far superior to standard bait mixes such as that used here [47,48]. Nevertheless, further research on optimal attractants is warranted.

Ultimately, monitoring the ongoing fate of threatened ground-dwelling marsupials like *Isoodon* and *Potorous* demands sampling locations where habitat for them is both optimal as well as sub-optimal. This is important to assess whether their status is changing in both space and time in relation to management efforts. To best tackle sub-optimal sites, it will be necessary to develop a modified monitoring approach that aims to improve the chance of detection, be it through using additional camera-traps, better attractants or a combination of both. Meanwhile, occupancy estimation across the landscape for these species will continue to be something less than ideal. For truly enigmatic species such as the *Perameles*, this situation may be more intractable without significant sampling breakthroughs.

To some extent it might be possible to explore the option of whether other, newer camera models are better suited to detecting the target species. For some ground-dwelling mammals differences in the effectiveness of camera-trap models can hinge on attributes such as detection zone [49] and it is also clear that different camera models can be variously seen and heard by animals [50]. Embarking down this path, however, demands clearly understanding how disparate different camera models are in their abilities and then being able to calibrate appropriately. This is inevitably core to any ongoing long-term monitoring program based on camera-trapping. However, it is not yet clear that finding superior camera models is any more important than resolving issues around ideal camera-trapping sampling frameworks.

## 5. Conclusions

Camera-traps are providing insights into occupancy rates of critical-weight-range marsupials across sites in south-eastern New South Wales, Australia. Our early efforts are demonstrating the persistence and relative stability of two endangered species, the southern brown bandicoot and long-nosed potoroo, as well as that of a third but more common species, the long-nosed bandicoot. That said, the detectability of these three species varies with habitat structure and complexity. The result of this is that where habitat is optimum for a given species, existing survey effort is adequate to be confident of detecting it. Outside of this optimal habitat, however, survey effort is not good enough to conclude with reliability that a species is in fact absent. Future survey and monitoring efforts for these species will need to take better account of these nuances and attempt to improve detection rates for these species through provision of additional camera-traps or novel lures and attractants. There may also be scope to explore the relative effectiveness of different camera models in detecting the target species. Pursuing that latter path, however, will require robust comparative trials with existing camera-trap units to enable proper calibration of ongoing monitoring efforts.

## Figures and Tables

**Figure 1 animals-09-00913-f001:**
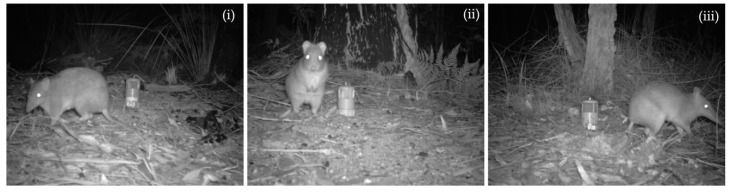
Night-time monochrome images captured by camera-traps of the three target species monitored in this study: (**i**) southern brown bandicoot *Isoodon obesulus*, (**ii**) long-nosed potoroo *Potorous tridactylus*, and (**iii**) long-nosed bandicoot *Perameles nasuta*. For scale, the containers used to hold bait also in the images are approximately 0.2 m high.

**Figure 2 animals-09-00913-f002:**
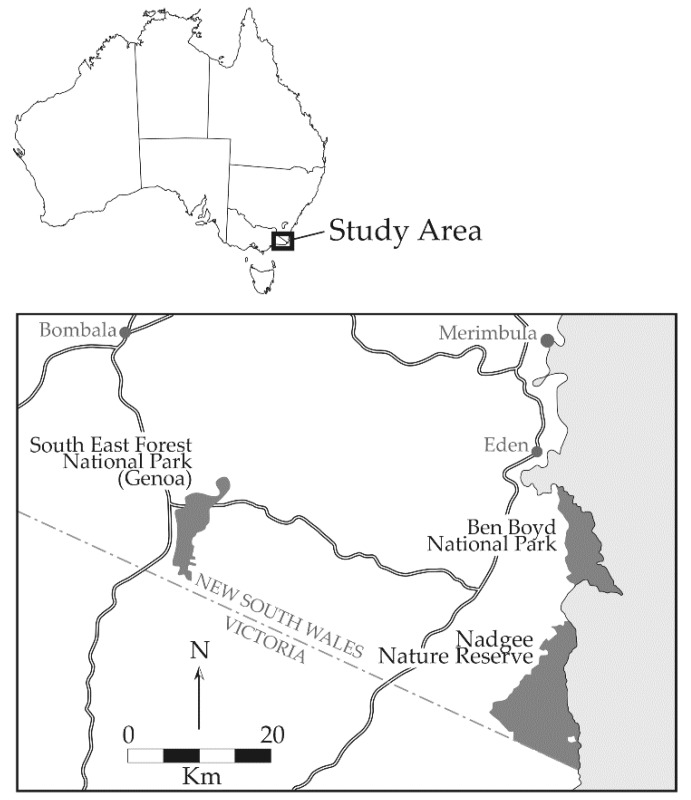
Map of the general study area in south-eastern New South Wales, Australia.

**Figure 3 animals-09-00913-f003:**
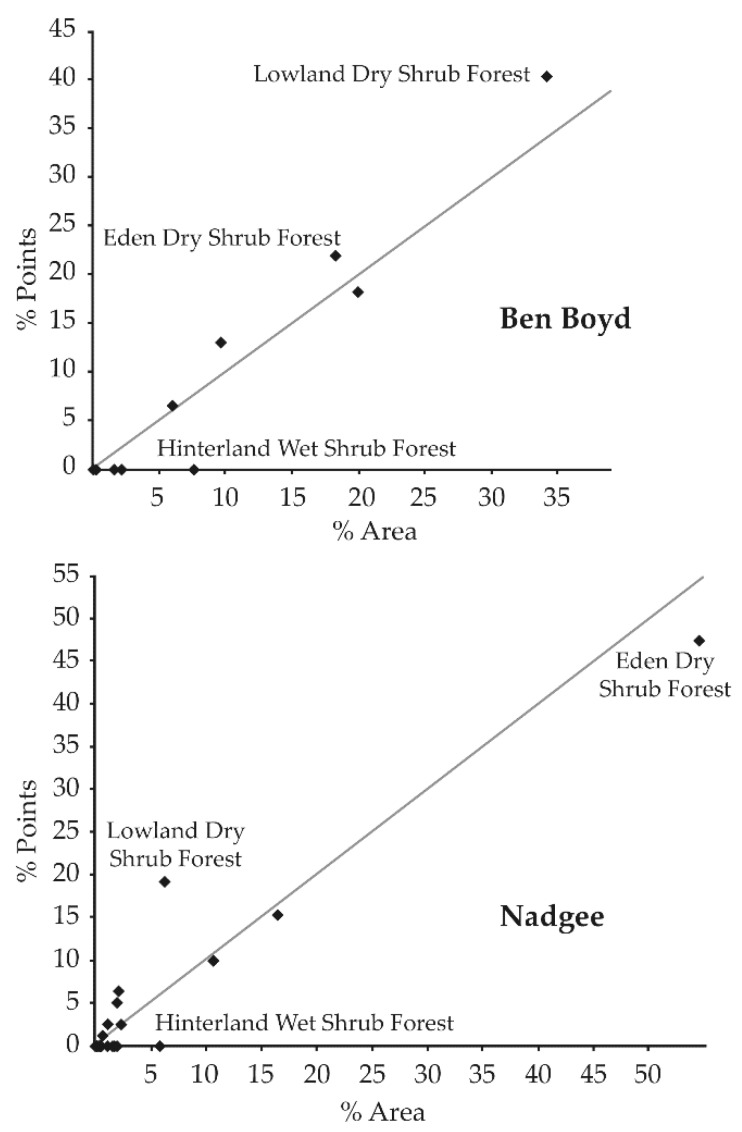
Distribution of camera-trap survey sites in relation to major vegetation classes across the study area. Vegetation classes proportionately under-sampled by initial desktop site allocation (indicated by written captions) were addressed in the field by re-allocating site locations.

**Figure 4 animals-09-00913-f004:**
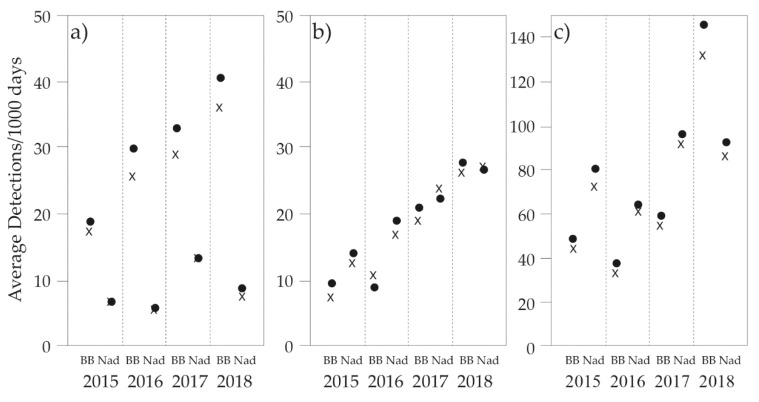
Average number of detections per thousand days for (**a**) *Isoodon obesulus*, (**b**) *Perameles nasuta*, and (**c**) *Potorous tridactylus* based on survey periods of 30-days (●) and 40-days (x) in Ben Boyd (BB) National Park and Nadgee (Nad) Nature Reserve.

**Figure 5 animals-09-00913-f005:**
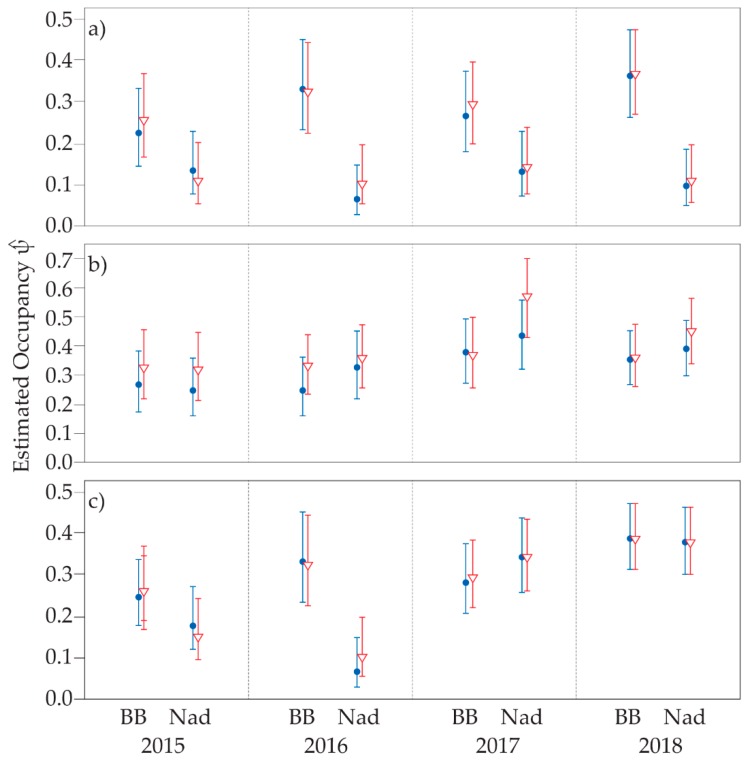
Single-season occupancy for (**a**) *Isoodon*, (**b**) *Perameles*, and (**c**) *Potorous* in Ben Boyd (BB) and Nadgee (Nad). Estimates are based on a 30-day survey period (blue) and 40-day survey period (red). Central mark represents the predicted estimate and bars represent the 95% confidence interval. Note that with the exception of occupancy estimates for *Perameles* in 2017, duration of camera-trap deployment had little effect on outcomes.

**Figure 6 animals-09-00913-f006:**
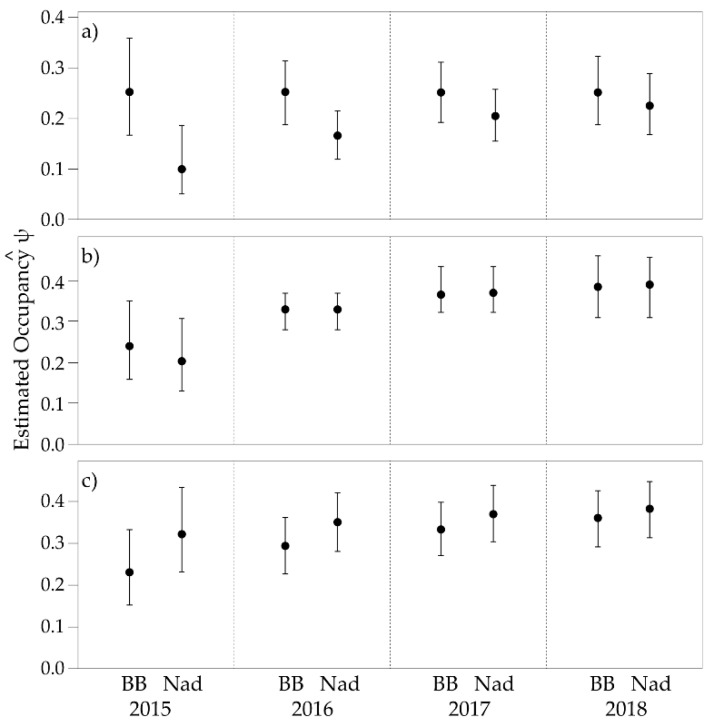
Multi-season occupancy estimates for (**a**) *Isoodon*, (**b**) *Perameles*, and (**c**) *Potorous* in Ben Boyd (BB) and Nadgee (Nad). Central mark represents the predicted estimate and bars represent the 95% confidence interval.

**Figure 7 animals-09-00913-f007:**
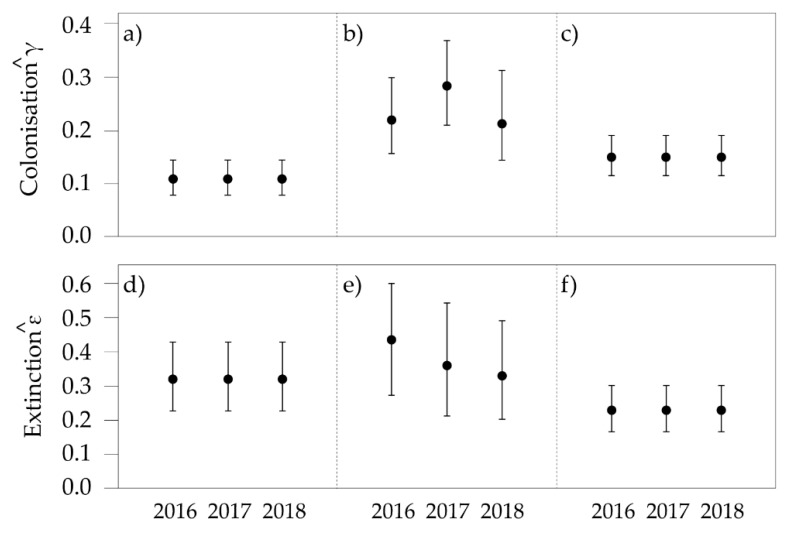
Estimated probabilities of colonisation (γ) and extinction (ε) for *Isoodon* (**a**,**d**), *Perameles* (**b**,**e**), and *Potorous* (**c**,**f**) during the 2016, 2017, and 2018 transitions. Central mark represents the predicted estimate and bars represent the 95% confidence interval.

**Figure 8 animals-09-00913-f008:**
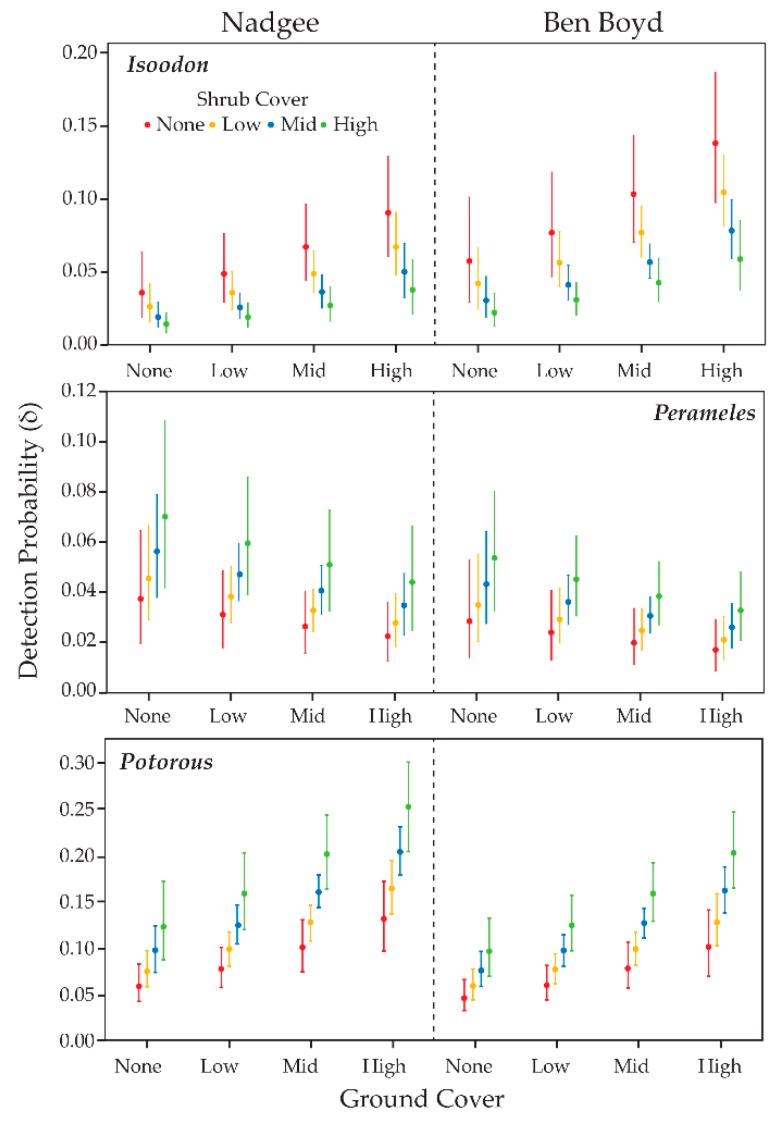
Estimated detection probability (*δ*) of *Isoodon*, *Perameles* and *Potorous* in Ben Boyd and Nadgee as a function of ground and shrub cover. As trends in each year were similar, only 2017 estimates are shown. Central dots represent the mean of the posterior distributions and bars represent 95% credible intervals.

**Figure 9 animals-09-00913-f009:**
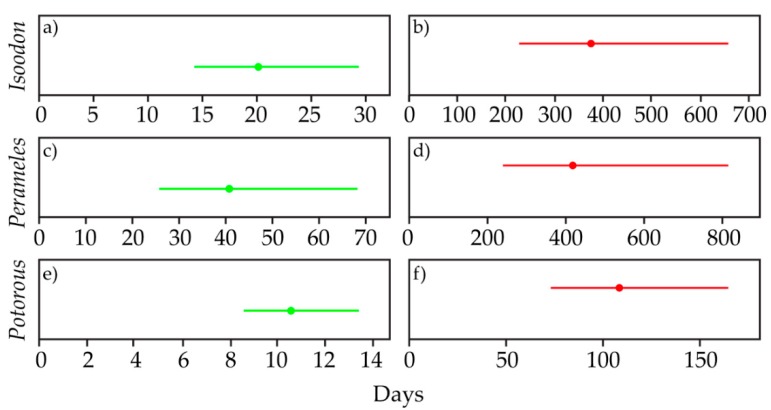
Number of days required to be 95% certain that non-detection of a target species (*Isoodon*, *Perameles* and *Potorous*) could be interpreted as its absence. Calculations are based on best case (**a**, **c**, and **e**; green; i.e., a favourable environment) and worst case (**b**, **d**, and **f**; red; i.e., an unfavorable detection environment) scenarios of the respective target species. Central mark represents the predicted estimate and bars represent the 95% confidence interval.

**Table 1 animals-09-00913-t001:** Habitat complexity scoring system used at camera trapping sites. Adapted from Newsome and Catling [29].

Habitat Attribute	Score			
	0	1	2	3
Tree Cover (>10 m) (%)	0	<30	30–70	>70
Shrub Cover (1–10 m) (%)	0	<30	30–70	>70
Ground (0–1 m) Cover (%)	0	<30	30–70	>70
Woody Debris (%)	0	<30	30–70	>70
Rocks (%)	0	<30	30–70	>70
Leaf Litter (%)	0	<30	30–70	>70
Moisture	Dry	Moist	Water Nearby	Water-Logged

**Table 2 animals-09-00913-t002:** Interpretation of the change in Akaike information criterion (ΔAIC) and deviance information criterion (ΔDIC) for assessing model support. Adapted from Burnham and Anderson [31] and McCarthy [33] (respectively).

ΔAIC/ΔDIC	Degree of Support
0–2	Substantial
4–7	Considerably Less
>10	Essentially None

**Table 3 animals-09-00913-t003:** Detection frequency of *Isoodon*, *Perameles* and *Potorous* during 2015–2018 for a 30-day (and 40-day) survey.

Year	*Isoodon*	*Perameles*	*Potorous*
2015	61 (78)	56 (65)	307 (374)
2016	84 (98)	67 (89)	243 (304)
2017	111 (136)	103 (136)	368 (374)
2018	124 (148)	137 (180)	599 (735)
Total	380 (460)	363 (470)	1517 (1880)

**Table 4 animals-09-00913-t004:** Single-season occupancy model selection results for *Isoodon*, *Perameles* and *Potorous* between 2015–2018, based on a 30-day survey period.

Year	Model	K	ΔAIC	AIC Weight
*Isoodon*				
2015	*ψ*(area) *p*(.)	3	0.00	0.71
	*ψ*(.) *p*(.)	2	1.79	0.29
2016	*ψ*(area) *p*(.)	3	0.00	0.99
	*ψ*(.) *p*(.)	2	15.22	<0.01
2017	*ψ*(area) *p*(.)	3	0.00	0.77
	*ψ*(.) *p*(.)	2	2.43	0.23
2018	*ψ*(area) *p*(.)	3	0.00	0.99
	*ψ*(.) *p*(.)	2	14.04	<0.01
*Perameles*				
2015	*ψ*(.) *p*(.)	2	0.00	0.67
	*ψ*(area) *p*(.)	3	1.40	0.33
2016	*ψ*(.) *p*(.)	2	0.00	0.57
	*ψ*(area) *p*(.)	3	0.57	0.43
2017	*ψ*(area) *p*(.)	3	0.00	0.52
	*ψ*(.) *p*(.)	2	0.17	0.48
2018	*ψ*(area) *p*(.)	3	0.00	0.58
	*ψ*(.) *p*(.)	2	0.67	0.42
*Potorous*				
2015	*ψ*(.) *p*(.)	2	0.00	0.52
	*ψ*(area) *p*(.)	3	0.16	0.48
2016	*ψ*(area) *p*(.)	3	0.00	0.84
	*ψ*(.) *p*(.)	2	3.35	0.16
2017	*ψ*(.) *p*(.)	2	0.00	0.54
	*ψ*(area) *p*(.)	3	0.34	0.46
2018	*ψ*(.) *p*(.)	2	0.00	0.71
	*ψ*(area) *p*(.)	3	1.77	0.29

‘K’ represents the number of parameters in the model and (.) denotes constant occupancy (*ψ*) or detection probability (*p*). ΔAIC is the change in Akaike information criterion. Only area (i.e., Ben Boyd or Nadgee) was included as a covariate in occupancy estimates.

**Table 5 animals-09-00913-t005:** Single-season occupancy model selection results for *Isoodon*, *Perameles* and *Potorous* between 2015–2018, based on a 40-day survey period.

Year	Model	K	ΔAIC	AIC Weight
*Isoodon*				
2015	*ψ*(area) *p*(.)	3	0.00	0.84
	*ψ*(.) *p*(.)	2	3.37	0.16
2016	*ψ*(area) *p*(.)	3	0.00	0.99
	*ψ*(.) *p*(.)	2	9.17	0.01
2017	*ψ*(area) *p*(.)	3	0.00	0.81
	*ψ*(.) *p*(.)	2	2.92	0.19
2018	*ψ*(area) *p*(.)	3	0.00	0.99
	*ψ*(.) *p*(.)	2	13.36	<0.01
*Perameles*				
2015	*ψ*(.) *p*(.)	2	0.00	0.72
	*ψ*(area) *p*(.)	3	1.89	0.28
2016	*ψ*(.) *p*(.)	2	0.00	0.64
	*ψ*(area) *p*(.)	3	1.16	0.36
2017	*ψ*(area) *p*(.)	3	0.00	1.00
	*ψ*(.) *p*(.)	2	57.55	0.00
2018	*ψ*(area) *p*(.)	3	0.00	1.00
	*ψ*(.) *p*(.)	2	124.84	0.00
*Potorous*				
2015	*ψ*(.) *p*(.)	2	0.00	0.54
	*ψ*(area) *p*(.)	3	0.35	0.46
2016	*ψ*(area) *p*(.)	3	0.00	0.70
	*ψ*(.) *p*(.)	2	1.71	0.30
2017	*ψ*(.) *p*(.)	2	0.00	0.52
	*ψ*(area) *p*(.)	3	0.17	0.48
2018	*ψ*(.) *p*(.)	2	0.00	0.71
	*ψ*(area) *p*(.)	3	1.77	0.29

‘K’ represents the number of parameters in the model and (.) denotes constant occupancy (*ψ*) or detection probability (*p*). ΔAIC is the change in Akaike information criterion. Only area (i.e., Ben Boyd or Nadgee) was included as a covariate in occupancy estimates.

**Table 6 animals-09-00913-t006:** Multi-season occupancy model selection results for *Isoodon*, *Perameles* and *Potorous*, based on 40-day survey period data.

Model	K	ΔAIC	AIC Weight
*Isoodon*			
*ψ*(area) γ(.) ε(.) *p*(.)	5	0.00	0.87
*ψ*(.) γ(.) ε(.) *p*(.)	4	4.26	0.10
*ψ*(area) γ(year) ε(year) *p*(.)	9	7.26	0.02
*ψ*(.) γ(year) ε(year) *p*(.)	8	52.14	0.01
*Perameles*			
*ψ*(.) γ(.) ε(.) *p*(.)	4	0.00	0.33
*ψ*(.) γ(year) ε(year) *p*(.)	8	0.53	0.25
*ψ*(area) γ(.) ε(.) *p*(.)	5	0.62	0.24
*ψ*(area) γ(year) ε(year) *p*(.)	9	1.11	0.19
*Potorous*			
*ψ*(.) γ(.) ε(.) *p*(.)	4	0.00	0.48
*ψ*(area) γ(.) ε(.) *p*(.)	5	0.23	0.43
*ψ*(.) γ(year) ε(year) *p*(.)	8	4.49	0.05
*ψ*(area) γ(year) ε(year) *p*(.)	9	4.71	0.04

‘K’ represents the number of parameters in the model. Models without covariates in occupancy (*ψ*), colonisation (γ), extinction (ε), or detection probability (*p*) are denoted by (.). ΔAIC is the change in Akaike information criterion. Area (i.e., Ben Boyd or Nadgee) and year were the only covariates considered.

**Table 7 animals-09-00913-t007:** Model scenarios used to estimate detection probability of *Isoodon, Perameles* and *Potorous* in Ben Boyd and Nadgee.

Model	DIC	ΔDIC	Iterations
***Isoodon***			
Baiting + Habitat + Year	2158	0	2500
Baiting + Habitat	2169	11	2500
Habitat	2178	20	2500
Baiting	2189	31	2500
Null	2199	41	2500
***Perameles***			
Baiting + Habitat + Year	2233	0	2500
Baiting + Habitat	2262	29	2500
Baiting	2263	30	2500
Habitat	2264	31	2500
Null	2265	32	2500
***Potorous***			
Baiting + Habitat + Year	5429	0	2500
Baiting + Habitat	5453	24	2500
Habitat	5466	37	2500
Baiting	5491	62	2500
Null	5499	70	2500

Predictors were whether baiting occurred at the site and habitat complexity scores of ground and shrub cover. Lower deviance information criterion (DIC) scores indicate the model with most support. Also shown is the number of iterations required for each model to reach convergence; burn-in was half iterations for all models.

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
