# Peer review of "Elucidating Patterns in the Occurrence of Threatened Ground-Dwelling Marsupials Using Camera-Traps"

_animals, 2019, doi:10.3390/ani9110913_

Round 1

Reviewer 1 Report

This paper is of high interest to camera trap researchers. It addresses the important issue of how to effectively survey for wildlife species. A range of different effects including species and vegetation are considered in order to determine effective sampling duration. The findings presented are of relevance to the Australian context and one wonders how more widely these findings apply. 

One suggestion for the authors is to include both images of species cohort studied and also the different vegetation covers considered. This would provide more context to the study and assist with its interpretation.

Overall, an excellent and interesting paper.

Reviewer 2 Report

Line 32: Check three critical-weight-range ground-dwelling marsupials – should there be a comma in there?

Line 63: Suggest expanding on the definition of critical-weight-range for international audience. Why is this weight range important – most extinctions? Also, this term comes up in the abstract and title – maybe doesn’t mean much for non-Australian reader.

Also – are they all marsupials? – if so might pay to keep referring to them as such rather than the generic term mammals.

Line 81: Suggest providing examples of background sign here – scats, tracks etc.

Line 108: Are you not also interested in the effect of control at one of the sites?

Line 113: When did the study take place?

Line 115: Can you quantify trap success 0.5% per 100 TN?

Line 162: These are old cameras – are the old cameras using truly invisible IR – what is the wavelength compared with modern cameras?

Line 189: I understand the need for spacing between cameras – do you have a reference for this spacing or is it based on HR size?

Figure 2: I am finding printouts of the Figures hard to read – can the resolution be bumped up? Also, on the Y-axis it looks like you have zero values – these need explanation. Additionally, you have only labelled three vegetation classes – are these the top three for each site?

Line 204: Did you randomise the camera site selections each year?

Line 210: OK you are assuming minimal effect of season and/or breeding? – however, could this also effect detection in the colder months?

Table 1: OK – you give Table 1 here for habitat complexity before the occupancy modelling section, but then you didn’t use as a covariate in the occupancy modelling.

This is where things get confusing for me. You analyse the data using occupancy modelling and then with Bayesian methods. For me you need to clearly split these approaches up and maybe discuss why you used two different approaches – strengths and weaknesses.

Line 256: You need the heading here to separate the different analyses.

Line 279: This time you used five models, but you only look at under and midstorey – how does this relate to Table 1. What is understorey – have you combined some values?

Line 294: Good explanation of Bayesian approaches – how did you do occupancy model – Presence software?

Line 327: You seem to be missing Section 3.3 – Multi Season occupancy write up which should come before Table 7.

Figure 4: Maybe highlight the impt. bit on the graph.

Figure 6: Has this been labelled correctly – shouldn’t it run from 2016-18 for the three years?

Line 376: This bit should be entitled as the Bayesian results section.

Line 392: You have a best- and worst-case scenario – you don’t mention this is the methods – how are these calculated and how should we interpret these?

Figure 7: This should be in the Results section and needs a line in the middle to separate the sites more clearly. Also why only results from 2017?

Figure 8: Hard to interpret as I don’t know what constitutes the worst-case position.

Line 467: “Dense low ground” – this is the understorey?

Line 491: Why do you use the current lure – is this backed up by any research or best guess?

Line 500: You mention additional camera traps – what about newer camera traps? Also given your results shown in Figure 7 – would you be better to target the optimal habitat for each species?

Conclusions: OK – all good – however, what about trap orientation and current setup – are there improvements that could be made there?
